# Comparative Study of Predominantly Daytime and Nighttime Lightning Occurrences and Their Impact on Ionospheric Disturbances

Louis Osei-Poku [1,2] , Long Tang [3] , Wu Chen [1,2,*] , Mingli Chen [4] and Akwasi Afrifa Acheampong [5]

1   Department of Land Surveying and Geo-Informatics, The Hong Kong Polytechnic University, Kowloon, Hong Kong; louis.oseipoku@connect.polyu.hk
2   Research Institute of Land and Space, The Hong Kong Polytechnic University, Kowloon, Hong Kong
3   School of Civil and Transportation Engineering, Guangdong University of Technology, Guangzhou 510006, China; ltang@gdut.edu.cn
4   Department of Building Environment and Energy Engineering, The Hong Kong Polytechnic University, Kowloon, Hong Kong; mingli.chen@polyu.edu.hk
5   Department of Geomatic Engineering, Kwame Nkrumah University of Science and Technology, Kumasi AK-448-4746, Ghana; aaacheampong.coe@knust.edu.gh
*   Correspondence: lswuchen@polyu.edu.hk

**Abstract:** Space weather events adversely impact the operations of Global Navigation Satellite Systems (GNSS). Understanding space weather mechanisms, interactions in the atmosphere, and the extent of their impact are useful in developing prediction and mitigation models. In this study, the hourly lightning occurrence and its impact on ionospheric disturbances, quantified using the Rate of Total electron content Index (ROTI), were assessed. The linear correlation between diurnal lightning activity and ROTI in the coastal region of southern China where lightning predominates in the daytime was initially negative contrary to a positive correlation in southern Africa where lighting predominates in the evening. After appreciating and applying the physical processes of gravity waves, electromagnetic waves and the Trimpi effect arising from lightning activity, and the time delay impact they have on the ionosphere, the negative correlation was overturned to a positive one using cross-correlation. GNSS has demonstrated its capability of revealing the impact lightning has on the ionosphere at various times of the day.

**Keywords:** GNSS; lightning; ROTI; gravity wave; daytime; nighttime

## 1. Introduction

The wide spatio-temporal advantages of Global Navigation Satellite Systems (GNSS) have made it an effective tool not only in positioning and navigation but also in studying and monitoring weather events. Geomagnetic and solar radiation, which are the key drivers of ionospheric plasma variations have been extensively observed using GNSS [1–4]. Tropospheric weather events such as cyclones [5–8], earthquakes [9–13], and/or man-made events such as rocket launches [14] and explosive bursts [15] have also been observed to cause plasma variations through the use of GNSS.

Thunderstorms/lightning, another troposphere weather event, has also garnered interest in the GNSS scientific community for adopting the advantages of GNSS to gain more insight into its activities. In recent GNSS lightning studies, Osei-Poku et al. [16] evaluated common total electron content (TEC) detrending techniques during lightning events. Rahmani et al. [17] probed the vertical coupling effect of a thunderstorm on the lower ionosphere. The ionospheric response to thunderstorms in West Africa has been reported by Ogunsua et al. [18], whereas Tang et al. [19] and Liu et al. [20] presented gravity waves resulting from thunderstorms. Blanc et al. [21] also showed gravity wave measurements in the ionosphere following thunderstorms in Europe. Lay [22] reported

acoustic wave activity above thunderstorms. Amin [23] also studied the hourly lightning activity and its effect on the ionosphere using GNSS in southern Africa.

The quest to harness GNSS for more insightful studies about lightning continues. As a recommendation in their work, Amin [23] suggested the use of more long-term data to study the correlation between hourly lightning activity and irregular ionospheric events as their study had a limited dataset with few case studies. The use of long-term data would provide more consistent, reliable, and insightful information In this regard, this current study uses four-year long-term data to study the relationship between the hourly lightning activity and the rate of the TEC index (ROTI) in the low-latitude coastal sea region of southern China. Similar techniques used in Amin [23] are deployed for an effective comparison. The results are compared to Amin [23] and those in other geographical regions. The initial results in this present study showed a negative linear correlation between hourly lightning activity and ROTI as opposed to the positive linear correlation in Amin [23]. One reason could be attributed to lightning interactions with the ionosphere in the different geographical regions. In the subsequent sections, the data and methods used are described. Further investigations are made to find and discuss the physical mechanisms underlying the differences in the results. The derived conclusions are then presented.

## 2. Data and Methods

### 2.1. Lightning Data

The lightning data is from a local lightning network in the coastal region of southern China. The network consists of about 17 Vaisala LS8000 sensors that provide the geolocation and source peak currents of lightning in the area [24]. The number of times a current is recorded is positively correlated with lightning activity [25]; a day with a lightning count greater than 10,000 is deemed a "lightning day" [16]. The data span is from 2014 to 2017.

### 2.2. GNSS Data

The local GNSS data were obtained from the Hong Kong Satellite Reference (HK SatRef). The GNSS receivers have a sampling rate of 30 s. Information on the network is given in the works of Ji et al. [26] and Kumar et al. [27]. More information on HK SatRef is available at the website of the Hong Kong Survey Department (https://www.geodetic.gov.hk/en/rinex/downv.aspx, (accessed on 14 June 2019)).

### 2.3. Method

#### 2.3.1. ROTI

ROTI defined by Pi et al. [28] as the root mean square of the TEC rate is used to characterize ionospheric irregularity. ROTI could be used as a proxy for the scintillation [29,30]. ROTI is computed from the GNSS data as follows. First, a 15° elevation cut-off angle was used to eliminate the multipath effect [6]. A geometry-free linear combination of pseudo- and carrier-phase signals was then used to compute slant TEC (STEC) at 30 s sampling intervals. STEC was converted to vertical TEC (VTEC) by applying a mapping function using Equation (1) below, where $R_e$ is the earth's radius, $\theta$ is the elevation angle at the ionospheric pierce point (IPP) of the signal–receiver path, and $h_i$ is the ionospheric single layer, approximated at 350 km.

$$\text{VTEC} = \sqrt{1 - \left(\frac{R_e \cos \theta}{R_e + h_i}\right)^2} * \text{STEC} \tag{1}$$

Finally, at five-minute intervals of the rate of TEC (ROT), ROTI was computed according to Equation (2), where ROT and ROTI are in TEC units (TECu: 1 TECu = $10^{16}$ e/m$^2$) and the notation $\langle \cdot \rangle$ is the averaging operation [31].

$$\text{ROTI} = \sqrt{<\text{ROT}^2> - <\text{ROT}>^2} \tag{2}$$

ROTI as shown in Equation (2) is usually for a single satellite-receiver pair. ROTI values exceeding 0.2 TECu are used to indicate that ionospheric scintillation has happened [32]. However, ROTI average (ROTI$_{avg}$) is the average value of ROTI over 30 min for all satellites received by a single station; following Oladipo et al. [33], a scintillation is deemed to occur for a ROTI$_{avg}$ value exceeding a threshold of 0.8 TECu. To effectively compare this study to that of Amin [23], ROTI$_{avg}$ is adopted.

### 2.3.2. Hourly Occurrence

For the diurnal hourly occurrence, the number of times lightning occurred and ROTI$_{avg}$ was greater than the threshold were accumulated within hourly intervals for all days in each year. A correlation is then looked for between the lightning and ROTI$_{avg}$ hourly occurrences.

### 2.3.3. Selection Criteria

To avoid the geomagnetic storm and solar radiation effects, only days with disturbance storm time (Dst) greater than −30 nT [34] and solar flux index (F10.7 index) less than 150 sfu (solar flux units, 1 sfu = $10^{-22}$ watt per square meter-hertz) [35] were selected. Table 1 shows the total number of days for the years 2014–2017 that had lightning activity only.

**Table 1.** Number of lightning days in each year void of geomagnetic storm and solar radiation effects.

| Year | Number of Lightning Days |
|------|--------------------------|
| 2014 | 85 |
| 2015 | 81 |
| 2016 | 108 |
| 2017 | 102 |

## 3. Results

According to Tang et al. [19], the stations of HK Satref are quite close, hence their observations are similar. Only the observations from one station (HKOH) are presented.

Figure 1 shows the ROTI$_{avg}$ from 2014 to 2017. In Figure 1, the highest values are mostly in the nighttime and between 0.07–0.2 TECu; this indicates that a scintillation is often a nighttime event. This is coherent with Tang et al. [31] and Ji et al. [26] who have shown that a greater percentage of ROTI lies between 0.02 and 0.05 TECu in the Hong Kong region. In this present study, the ROTI$_{avg}$ threshold is set at 0.075 TECu instead of 0.8 TECu, unlike Oladipo et al. [33] but the same as Nishioka et al. [36], who suggested a scintillation threshold of 0.075 TECu in the Asian region. More so, the years 2014–2017 are at the declining phase of the 24th solar cycle [37] where scintillation values are low compared to the high solar active years of 2002 and 2012 in Oladipo et al. [33] and Amin [23], respectively. Also, Jacobsen [38] and Liu et al. [39] have demonstrated that GNSS receiver types, configurations, and sampling rates influence ROTI values. These differences in ROTI values (thresholds) arising from the technique used, GNSS receiver configurations, and geographic location should be important factors to consider when developing regional and global models.

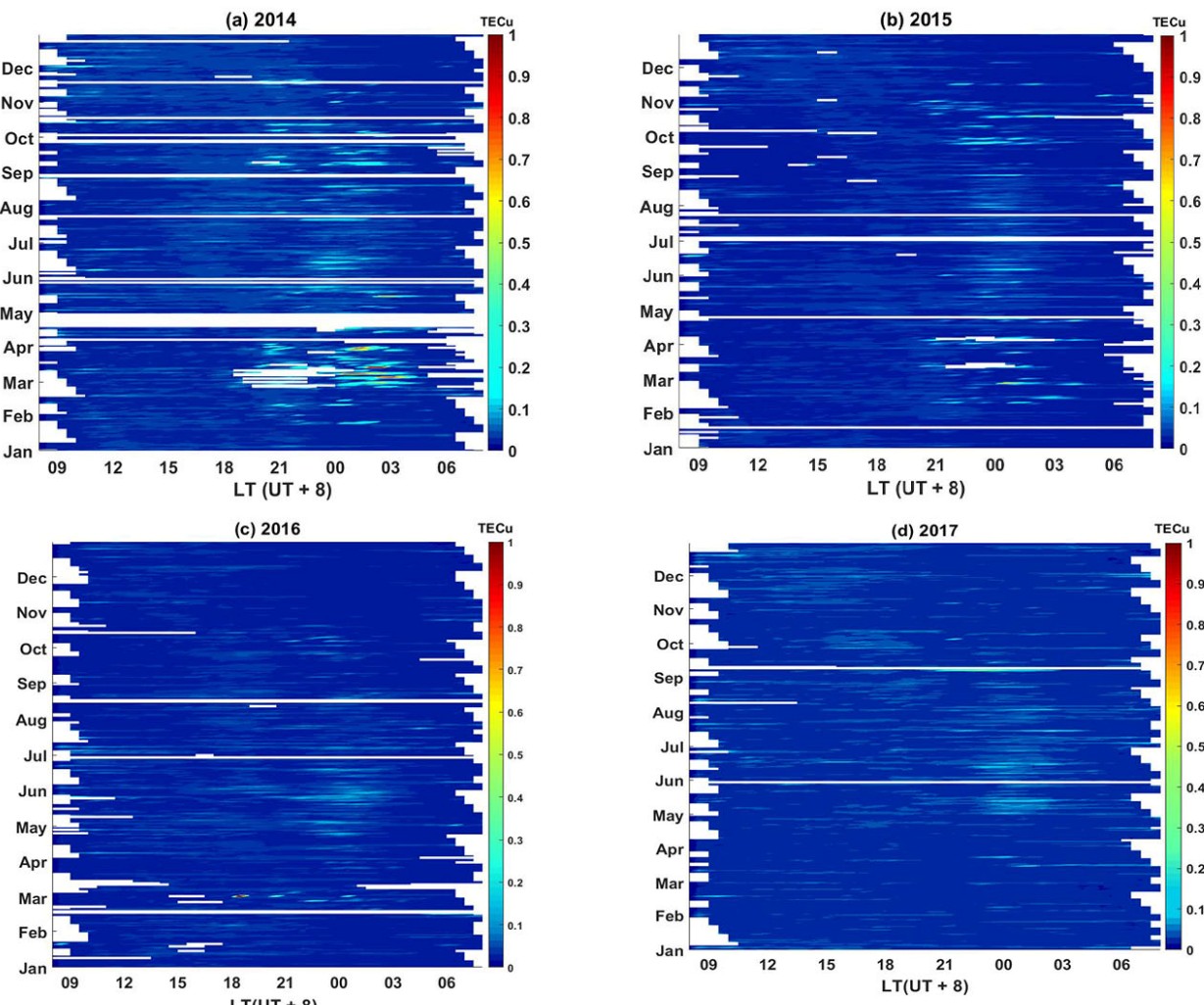

**Figure 1.** $ROTI_{avg}$ for the years 2014–2017. The abscissa axis shows the local time (LT: UT + 8) in hours. The ordinate axis shows the months of the year. The color bar shows the $ROTI_{avg}$ values in TECu.

On the diurnal hourly occurrence, the number of times lightning occurred and the $ROTI_{avg}$ was greater than 0.075 TECu were accumulated within hourly intervals for all days in each year. Panels a to d in Figure 2 show the annual hourly diurnal $ROTI_{avg}$ occurrence against that of the lightning occurrence. All years showed a similar trend. The trend reveals that the lighting occurrence peaked before that of $ROTI_{avg}$. This resulted in negative linear correlation values of $-0.364$, $-0.41$, $-0.371$, and $-0.421$ with significant values (*p*-value) of 0.05, 0.04, 0.05, and 0.04 at a confidence interval of 95% ($\alpha = 0.05$) in chronological order from 2014 to 2017 as seen in panels e to h in the right column in Figure 2. The *p*-values indicate that despite being negative, the linear correlation is statistically significant. This implies that as lightning increases, ROTI decreases and vice versa. These are contrary to observations in Amin [23] where both lightning and $ROTI_{avg}$ peaked at the same time in the evening resulting in positive linear correlation values. As the objective of this study is to find the relationship between lightning and ROTI, reasons are discussed to explain the differences between this observation and that of [23].

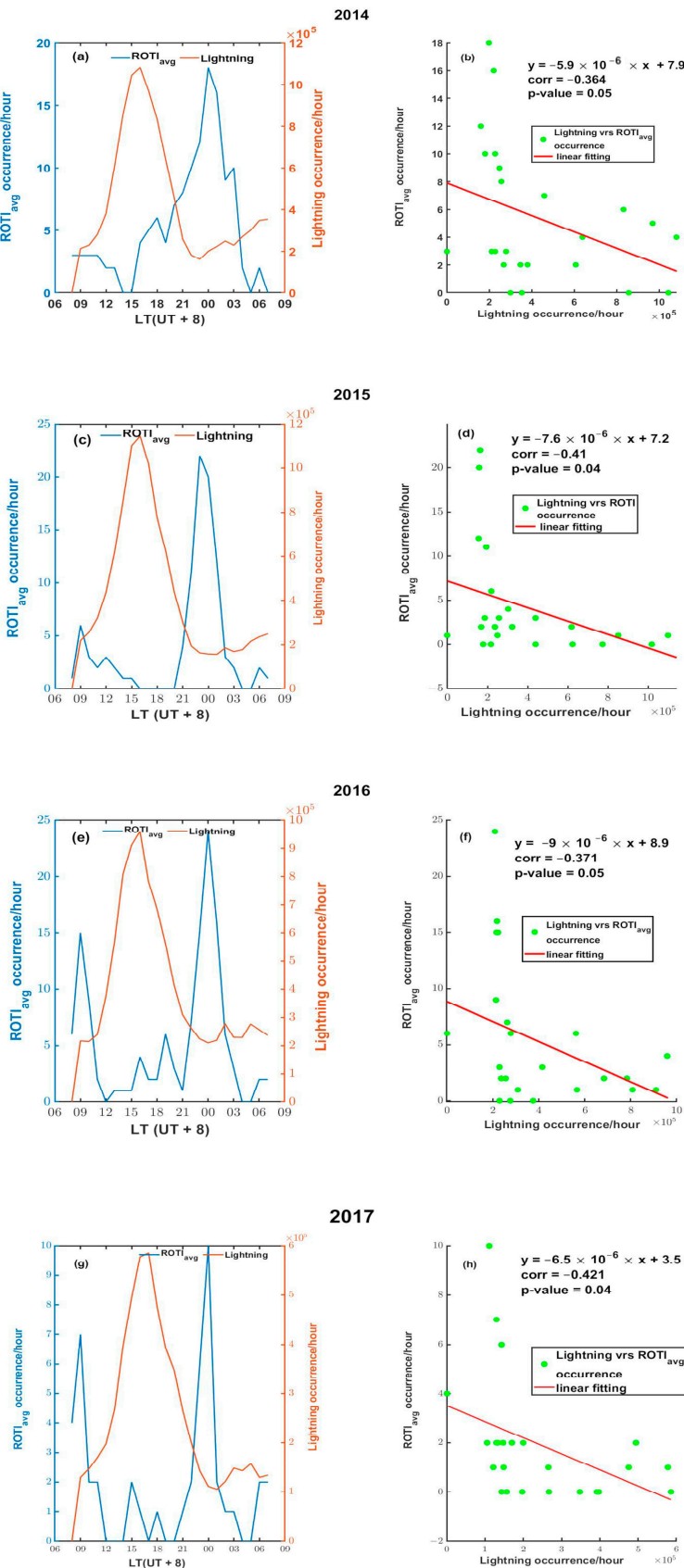

**Figure 2.** Panels in the left column (**a**–**d**) show the diurnal hourly occurrence of ROTI_avg and lightning. Panels in the right column (**e**–**h**) are scatter diagrams showing the linear correlation between the diurnal hourly occurrence of lightning and ROTI_avg for the respective years.

## 4. Discussions

The objective of this work is to appreciate and understand the relationship between hourly lightning activity and ionospheric irregularities (ROTI) through the use of long-term data. The initial results seen in Figure 2 show a negative linear correlation contrary to the positive linear correlation reported in Amin [23]. One reason for the observed differences in linear correlation could be attributed to the interaction between local variations in mesoscale and the topographic nature of the geographic region [40,41] resulting in different diurnal trends of lightning. That is, different regions tend to respond to lightning differently. From the observed data and results in other studies, there seems to be a common trend in the diurnal pattern of lightning. Places in the northern hemisphere seem to have a common pattern. Likewise, places in the southern hemisphere. For instance, Figure 5 in Williams et al. [42] shows a similar pattern of diurnal lightning as observed in this current study, and their Figure 6 is similar to that of Figure 5.13 of Amin [23]. In Figure 5 of Williams et al. [42], Australia, which is in the southern hemisphere, shows a broad peak of lightning from noon, which lasts until about 23:00 local time (LT). This pattern is similar to that of South Africa, also in the southern hemisphere. Brazil also shows a similar trend as seen in Pinto et al. [40]. In Figure 6 of Williams et al. [42], French Guyana, which is in the northern hemisphere and in South America, exhibits a lightning peak of around 15 LT, which lasts for about 3 h (till 18 LT). This pattern is similar to that of Hong Kong. Other similar trends are observed in the Indian region [43] and the United States [41,44], all in the northern hemisphere. The broad width of the lightning peak observed in the southern hemisphere regions should cover their peak time of $ROTI_{avg}$, which happens in the evening (mostly between 20 and 23 h). Statistically, lightning and $ROTI_{avg}$ having similar trends and with their peaks coinciding at the same time stands a high chance of giving a positive linear correlation value, whereas peaks that occur far apart at different times would give a negative linear correlation value.

Correlation may not necessarily mean causation. Possible mechanisms influencing the diurnal correlations in this work and that of Amin [23] are the lightning activity interactions with the ionosphere through lightning-related current and electrical discharges [45,46] and processes of gravity and electromagnetic wave [24,47–49].

A thunderstorm with lightning activity may affect the electron density in the lower ionosphere (particularly the D-layer) through two basic physical processes: the gravity wave (GW) due to the convection and thunder [47] and the electromagnetic wave (EM) due to the lightning [24]. The GW will generate a disturbance of the mesosphere (60–90 km) and the lower thermosphere (above 90 km), thus of the neutral atmosphere, which drags electrons from the ionosphere. Although the GW is a transversal wave, any changes in its amplitude generated from a lightning discharge will perform as a longitudinal wave that propagates upwards at the speed of the order of sound in air. The higher the air density, the stronger and faster the GW. Therefore, the GW due to the lightning and thunder and the thunderstorm convection disturbance would be stronger and faster as it propagates in the lower (higher air density) D-layer during the daytime than it does in the higher (lower air density) D-layer during the nighttime. The GW due to the lightning and thunder and the thunderstorm convection disturbance on ROTI lags the lightning activity due to its very low propagation speed at a high altitude. Theoretical studies have shown that the GW needs about one to several hours to propagate to higher altitudes depending on speed and period [50]. Using the wave model of Row [51] and Francis [52], which assumes that the GW propagates upwards through the atmosphere, Taylor et al. [53] found that the GW from a thunderstorm took about six hours before reaching the airglow layers (80 km above ground surface), which implies a vertical velocity of about 5 ms$^{-1}$. In Figure 2, the time delay between lightning and $ROTI_{avg}$ is about 5–7 h, similar to that observed by [53]. At an altitude of 350 km and a time delay of 5–7 h, the vertical velocity of the GW would be approximately 10–15 ms$^{-1}$. These observations show that the average vertical velocity of the GW is about 5–15 ms$^{-1}$ confirming the low propagation speed of the GW. Also, the GW is predominant in the horizontal component which takes a relatively

longer time to reach higher atmosphere heights [19,20,54]. This could best account for the difference in peak times of lightning and the ROTI pattern in this study, where lightning is predominantly in the afternoon. Again, Lay [22] made an observation where lightning occurring predominantly in the daytime (14–16 LT) had its related ionospheric disturbances around 00–02 LT similar to the observations in the left panels in Figure 2. Thus, lightning that occurs predominantly in the afternoon sees the ionospheric disturbance some hours later leading to an initial negative linear correlation.

Lightning-generated electromagnetic waves (EM) on the other hand propagate at the speed of light and reach the lower ionosphere almost immediately after the lightning activity has taken place. The strength of the impact of the EM on ROTI depends on the density of the electrons present. The higher the electron density, the stronger the impact. Theoretical simulations have shown that the EM could lead to a reduction in electron density at lower altitudes and an increase in electron density at higher altitudes in an ionospheric D-layer due to Joule heating effects [55–57], which have also been proven by lightning sferics observations [24,58,59]. Lightning sferics also produce very-low-frequency (VLF) discharges that contribute to the Trimpi effect. The Trimpi effect refers to transient perturbations caused by electron precipitation on sub ionospheric propagating waves [60]. The perturbations associated with sferics take only about 0.6 s to begin. Also, the Trimpi effect is only observed in nighttime ionospheric conditions and not daytime [61,62]. Therefore, the lightning-generated EM and the Trimpi effect would have a stronger impact on ROTI during the nighttime than during the daytime. The appearance of the impact of the EM and Trimpi effect on ROTI is almost on par with the time the lightning activity happened. Thus, lightning that occurs predominantly in the evening sees an ionospheric disturbance almost instantly leading to a positive linear correlation.

To further illustrate the time of lightning dominance and the time lags of its associated ionospheric disturbance, two diurnal zone divisions are made. That is, 07–17 and 18–06 LT to represent the daytime and evening time zones, respectively [30]. In Figure 2, 2014–2017 have similar lightning and ROTI$_{avg}$ trends. Some days are selected to illustrate this further as most of the individual days have similar observations. The assessment is done by finding the cross-correlation between lightning and ROTI instead of the linear correlation. Cross-correlation studies different variables to identify their similarity and draws characteristics relative to each other based on time to derive new information. Pseudorandom noise codes (PRN) from the individual satellite-receiver pairs available at the time ROTI lags the lightning are presented to provide extra evidence of ionospheric disturbances. The TEC of the PRN is detrended using the Savitzky–Golay filter of order 6 and window length of 120 min [16]. Detrended TEC (DTEC) is filtered with a bandpass of frequencies between 1 and 2.8 MHz and 4.2 and 8.2 MHz to derive ionospheric gravity (IGW) and ionospheric acoustic (IAW) waves, respectively, from lightning [17,63]. A filtered DTEC amplitude above 0.08 TECu and 0.025 TECu [63] and ROTI greater than 0.2 TECu signify the presence of IGW, IAW, and ionospheric disturbances, respectively. The selected days for the daytime and evening are presented in Figures 3 and 4, respectively. The left panels of the top rows for each day in Figures 3 and 4 show the ROTI$_{avg}$ and lightning counts. The right panels of the top rows show the cross-correlation between the ROTI$_{avg}$ and lightning counts from the left panel. The bottom rows show the DTEC, IGW, IAW, and ROTI of the PRN, which was available at the time ROTI lags the lightning. The magenta lines are the thresholds of IGW, IAW, and ROTI.

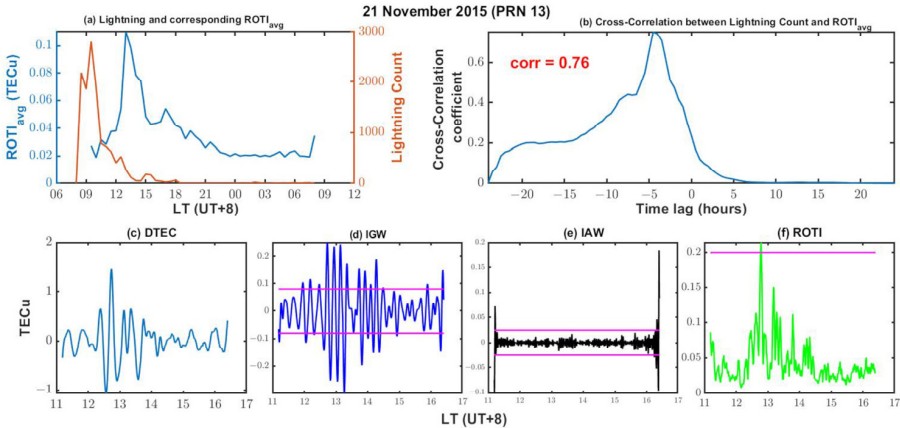

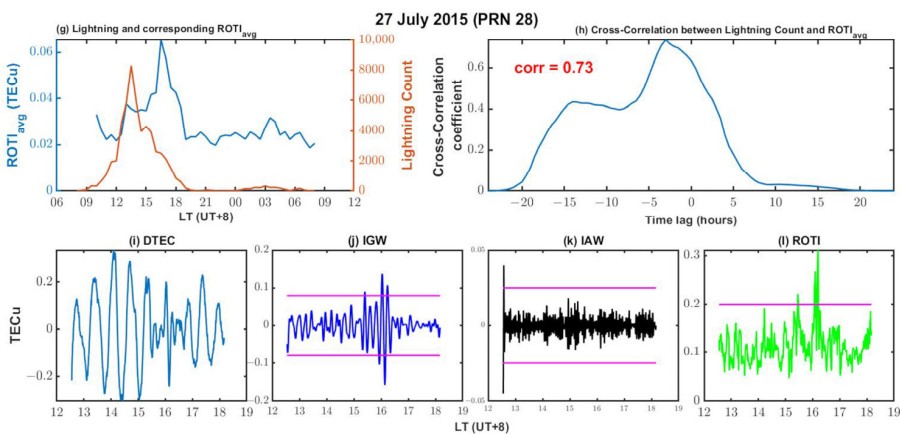

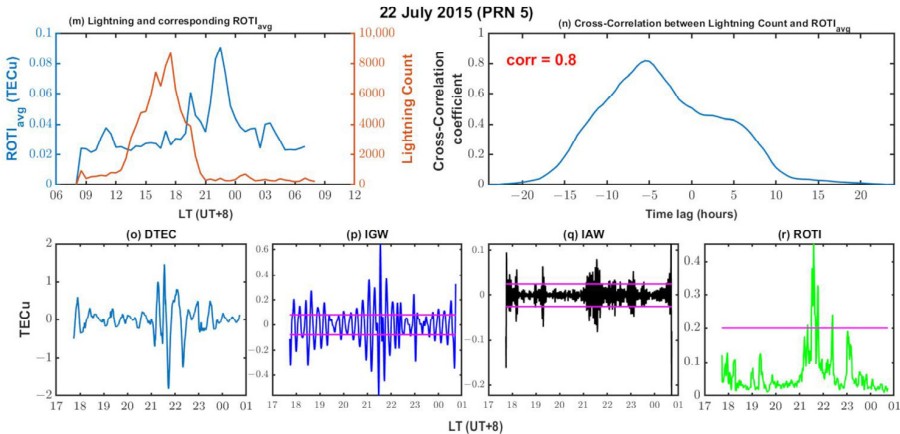

**Figure 3.** Some selected days (21 November 2015, 27 July 2015, 22 July 2015) where lightning was predominantly in the daytime (07–17 local time) and its cross-correlation with ROTI$_{avg}$ (**top rows of each day**). The associated ionospheric disturbances are also indicated by DTEC, IGW, IAW, and ROTI (**bottom rows of each day**) by satellites that passed some hours after the lightning activity. The magenta lines show the threshold of ±0.08 TECu, ±0.025 TECu and 0.2 TECu for IGW, IAW and ROTI respectively.

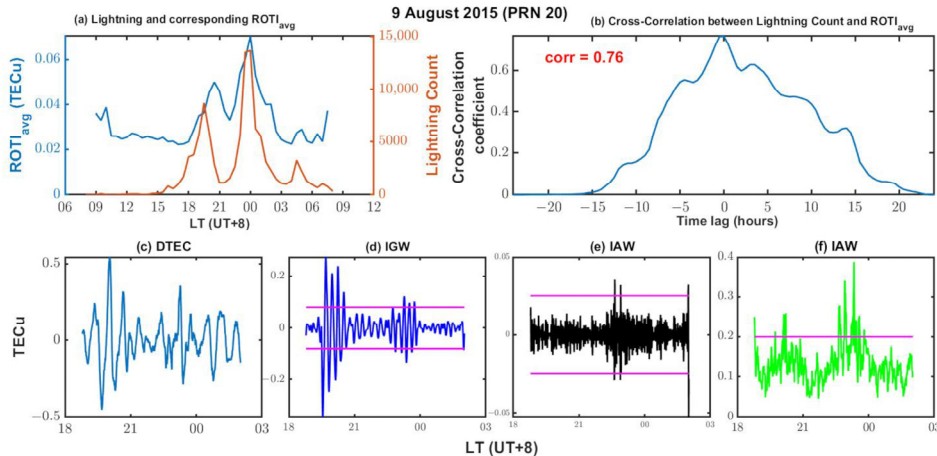

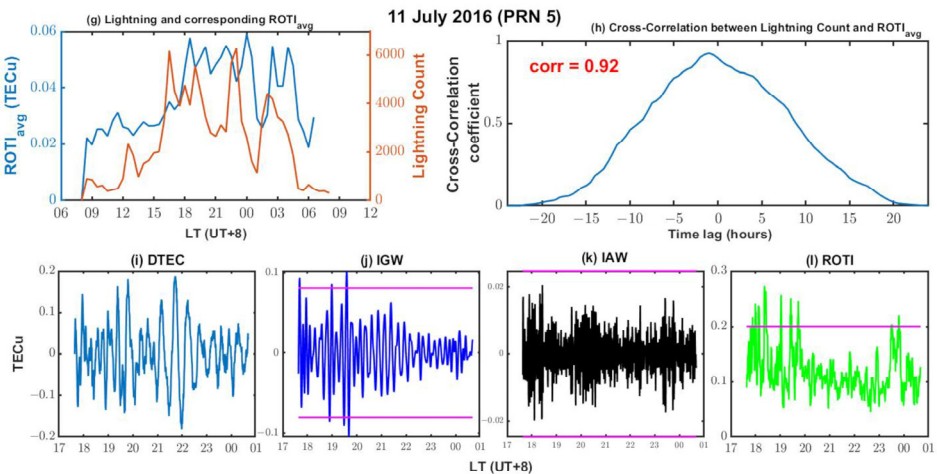

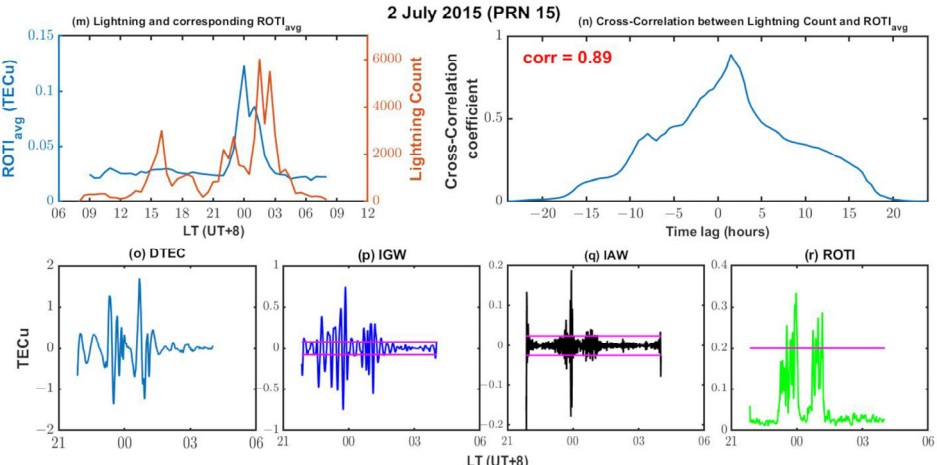

**Figure 4.** Some selected days (9 August 2015, 11 July 2016, 2 July 2015) where lightning was predominantly in the evening (18–06 local time) and its cross-correlation with ROTI$_{avg}$ (**top rows of each day**). The associated ionospheric disturbances indicated by DTEC, IGW, IAW, and ROTI (**bottom rows of each day**) by satellites that passed at the time of lightning activity. The magenta lines show the threshold of $\pm 0.08$ TECu, $\pm 0.025$ TECu and 0.2 TECu for IGW, IAW and ROTI respectively.

Figure 3 shows the days with lightning predominantly in the daytime. In panels b, h, and n in Figure 3, $ROTI_{avg}$ is seen to have a negative time lag of about 4 h to the lightning activity but is strongly positively correlated. The time of the disturbances as seen in the PRNs in the bottom rows confirms the delay. The negative time lags; however, positive coefficients indicate that the disturbances are associated with lightning, which can be attributed to the delayed impact of the gravity wave mechanism from the predominantly daytime lightning on the ionosphere.

Figure 4 shows the days with lightning predominantly in the evening. From panels b, h, and n in Figure 4, $ROTI_{avg}$ is seen to have a time lag of zero to a few minutes to the lightning activity and with strong positive correlation coefficients. The time of the disturbances as seen in the PRNs in the bottom rows is on par with the lightning activity. The time lag of zero to a few minutes and the positive coefficients indicate that the disturbances are associated with lightning, which can be explained by the almost immediate impact of the electromagnetic wave mechanism and the Trimpi effect resulting from the predominantly nighttime lightning on the ionosphere.

Following the mechanisms of the gravity wave, electromagnetic wave, and the Trimpi effect as discussed above, cross-correlation of the lightning and $ROTI_{avg}$ counts from the left panels in Figure 2 yielded positive correlation values of 0.89, 0.82, 0.72, and 0.61 for 2014 to 2017, respectively as shown in Figure 5. The values are similar to 0.86 reported by Amin [23]. This shows that the diurnal hourly lightning activity and associated $ROTI_{avg}$ have a positive correlation.

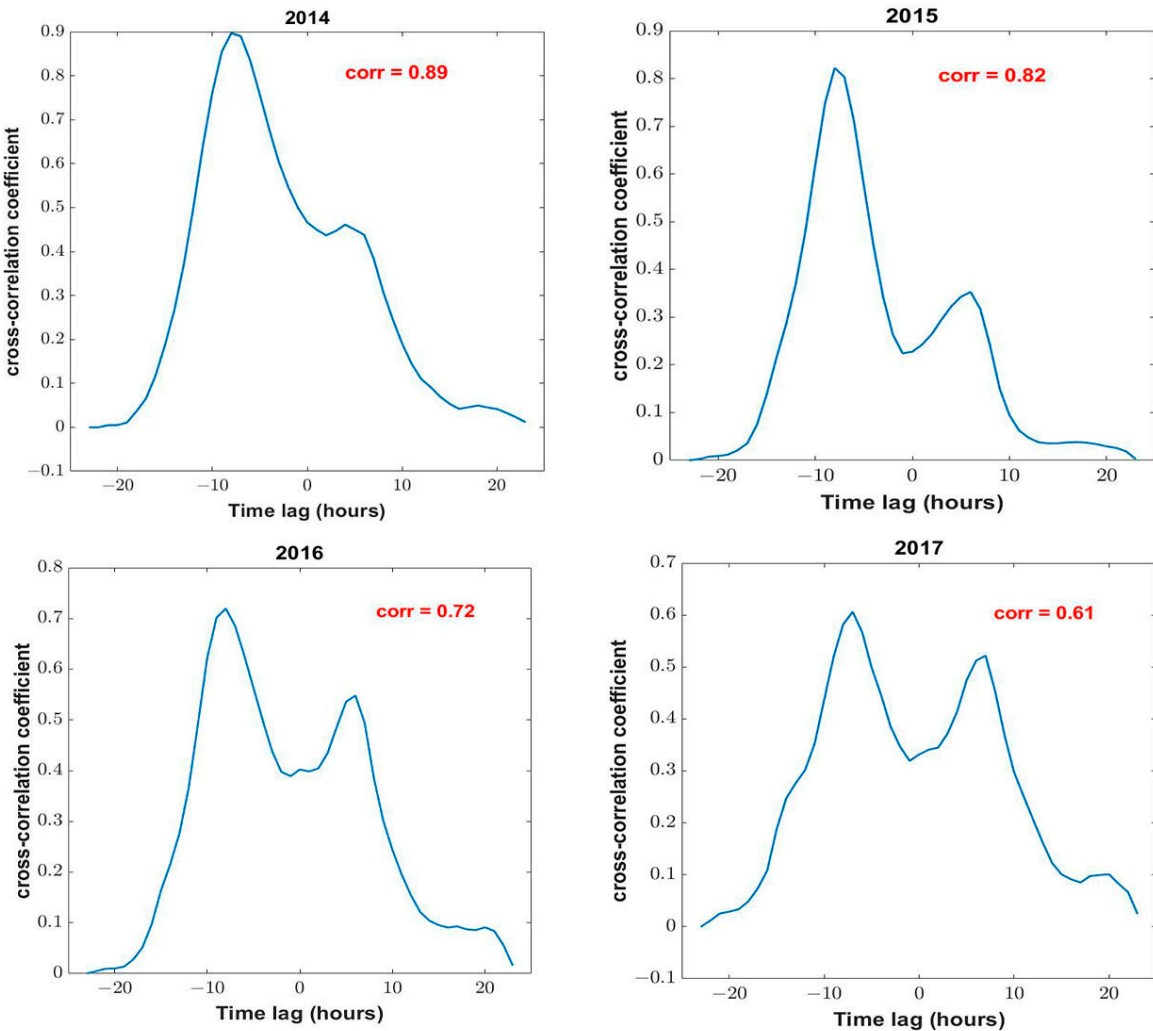

**Figure 5.** Cross-correlation of lightning and $ROTI_{avg}$ counts from left panels in Figure 2.

The observations in Figures 3–5 confirm that ionospheric disturbances from lightning occurring in the afternoon take a relatively longer time to manifest, whereas those due to lightning occurring in the evening are almost immediate. Using cross-correlation provides a better means of accessing the relationship between lightning activity and its associated ROTI at various times of the day. The cross-correlation coefficient is statistically significant when its absolute value is greater than the expected cross-correlation coefficient (ECCF), which is computed as [64].

$$ECCF = \frac{2}{\sqrt{n - |k|}} \tag{3}$$

where n is the total number of observations and k is the number of observations between zero and the point of the maximum cross-correlation value on the lag axis. Negative and positive k are to the left and right of zero on the lag axis, respectively.

Table 2 is a summary of the cross-correlation coefficients of days and years in Figures 3–5 and their ECCF. The observations of the days in Figures 3 and 4 are at 30 min intervals, whereas those of the years in Figure 5 are at hourly intervals. The cross-correlation coefficients being greater than the ECCFs show that lightning activity is significantly positively correlated with its associated ROTI regardless of the time of day the lightning was predominant.

**Table 2.** Cross-correlation coefficient of days and years in Figures 3–5 and their ECCF.

| | | Cross-Correlation Coefficient | n | k | ECCF |
|---|---|---|---|---|---|
| Days in Figure 3 | 21 November 2015 | 0.76 | 48 | −9 | 0.32 |
| | 27 July 2015 | 0.76 | 48 | −6 | 0.31 |
| | 22 July 2015 | 0.8 | 48 | −11 | 0.33 |
| Days in Figure 4 | 9 August 2015 | 0.76 | 48 | 0 | 0.3 |
| | 2 July 2015 | 0.89 | 48 | −1 | 0.29 |
| | 11 July 2016 | 0.92 | 48 | −1 | 0.29 |
| Years in Figure 5 | 2014 | 0.89 | 24 | −8 | 0.5 |
| | 2015 | 0.82 | 24 | −8 | 0.5 |
| | 2016 | 0.71 | 24 | −8 | 0.5 |
| | 2017 | 0.6 | 24 | −7 | 0.4 |

## 5. Conclusions

In this work, a correlational study between hourly lightning activity and ROTI was carried out using long-term data. The data were from 2014 to 2017 with the study area being Hong Kong. The linear correlation between the hourly lightning activity and ROTI was negative compared to a similar study in southern Africa, where the correlation was positive. The initial differences in linear correlation values between this present study and that in southern Africa are attributed to the time differences at which lightning and ROTI peaked. Although lightning peaked in the daytime in Hong Kong, its associated ROTI peaked in the evening. For southern Africa, both lightning and ROTI peaked at the same time in the evening. Probing further, this current study found that the ROTI resulting from lightning predominantly in the daytime lagged due to slow propagation of gravity wave mechanisms but did not lag when lightning was predominantly in the evening due to electromagnetic waves and the Trimpi mechanisms. The temporal discrepancy could be explained by either or both mechanisms and it would require simulations to confirm their likelihood. Using datasets obtained from radio occultation missions such as COSMIC would be effective for quantifying these mechanisms and their perturbations at the various layers of the ionosphere. Unlike the linear correlation that showed that predominantly daytime lightning is negatively correlated with its ROTI, cross-correlation offered a better means to access the lightning activity–ROTI relationship. Cross-correlation revealed the time delays and showed that the lightning activity and its associated ROTI are positively correlated, which can be explained by the physical mechanisms. The results and observations have shown that GNSS can reveal the impact lightning activity has on the

ionosphere at various times of the day. The models for ionospheric scintillation simulations, prediction, and forecasting purposes based on lightning activities should consider and incorporate this observation when being developed.

**Author Contributions:** Conceptualization and research design were done by L.O.-P. and W.C. L.O.-P., L.T., M.C. and A.A.A. contributed to data processing and analysis. W.C. and M.C. supervised the work. Writing and editing of the manuscript were done by all authors. All authors have read and agreed to the published version of the manuscript.

**Funding:** This research was funded by the University Grants Committee of Hong Kong under the scheme Research Impact Fund on the project R5009-21, and the Research Institute of Land and Space, Hong Kong Polytechnic University, and the National Natural Science Foundation of China (Grant number 41804021).

**Data Availability Statement:** The Dst-index data (IAGA 2002-like format) can be obtained from the Data Analysis Center for Geomagnetism and Space Magnetism, Kyoto University, operating WDC for Geomagnetism, Kyoto (http://wdc.kugi.kyoto-u.ac.jp/dstae/index.html). The Bz component of the interplanetary magnetic field was obtained from the GSFC/SPDF OMNIWeb interface at http://omniweb.gsfc.nasa.gov. The GNSS data can be obtained from Hong Kong SatRef of the Lands Department of the Hong Kong Government (https://www.geodetic.gov.hk/en/rinex/downv.aspx). All data center websites were accessed on 14 June 2019.

**Acknowledgments:** The research institutes and government departments where data was obtained for this study are duly acknowledged for their continuous support of scientific studies. The authors also thank the anonymous reviewers for their useful and insightful comments for improving the manuscript.

**Conflicts of Interest:** The authors declare no conflict of interest.

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
