# Peer review of "Comparative Study of Predominantly Daytime and Nighttime Lightning Occurrences and Their Impact on Ionospheric Disturbances"

_remotesensing, doi:10.3390/rs14133209_

Round 1

Reviewer 1 Report

The manuscript “Comparative study of Dominant Daytime and Nighttime

Lightning occurrence and their impacts on Ionosphere Disturbances” is

dedicated rather important studying of the ionosphere response to the

tropospheric lightnings. The phenomena is valuable for modeling of upward

impact to the ionosphere and can be taking in to account in the space

weather forecasting. Usage the GNSS stations as a indicator of response is

very interesting because the devices are widely distributed and in case of

positive results may be valuable meaning for the forecasting.

Author Response

Dear Reviewer,

Thank you for your valuable inputs in improving the manuscript and recommending for it to be published.

Reviewer 2 Report

I do not feel that the authors improved their paper deeply.

1) I indicated “The authors introduced “new ROTI” that differs significantly from those suggested by Pi et al. Pi et al used 30-sec slant TEC data, while here 5-min vertical TEC data is used. I do not think that at such a low resolution this “new ROTI” would correlate with ionospheric scintillation (as in [21,22]) due to different time scales. According to Jacobsen, even different sampling rates could result in different effects (DOI: 10.1051/swsc/2014031).”

The authors responded

The “New ROTI” as you would state it is adopted from the study of Amin  (Influence of lightning on electron density variation in the ionosphere using WWLLN lightning data and GPS data. Masters Thesis, University of Cape Town, 2015.) which is the main reference of this current study to allow effective comparison.

“New ROTI”  is the average of ROTI (originally discussed by Pi et al) over 30 min interval for a satellite and then average over all satellites. This concept was introduced by Oladipo et al.  (Equatorial ionospheric irregularities using GPS TEC derived index. Journal of Atmospheric and Solar-Terrestrial Physics 2013, 92, 78-82, doi:10.1016/j.jastp.2012.09.019.)

 The view of Jacobsen on sampling rate has been incorporated to enrich the discussion of the manuscript as suggested

The response is irrelevant. “New ROTI” is not an averaged ROTI. When you calculate a derivative, <dI/dt> is not d<I>/dt. It is also insignificant, that such a name was used in Master Thesis or other paper. Use other name for a new index. Moreover, you can not use those relationships for ROTI and scintillations for “New ROTI” and scintillations – the correlation should be so different, because of different time scales. 30-sec dI/dt reflects scintillations, while 5-min dI/dt reflects much higher irregularities (much more than Fresnel size).

2) I indicated: Figure 2 is so unconvincing. I would say about no correlation. There are different ROTI occurrences at low lightning occurrence (which occur more often) and constant ROTI at the different (higher) lightning occurrences. From this figure, I would conclude that lightning does not influence ROTI.

The authors response:

The non-correlation gives us reasons to investigate more the mechanisms of lightning and how they influence the ionosphere. By just looking at the figures, general statistics, and hindsight knowledge show that positive correlation would be recorded in Amin et al (Fig 5.13). This was one key reason for their conclusion that lightning is positively correlated with ROTI. In this current study, the figures (Fig 2) say otherwise. The further investigation into the differences showed that depending on the time of day of dominant lightning, ROTI impact would be almost immediate or delayed for some hours. The mechanisms have been described in the Discussion section.

Because 2014-2017 have similar trends (see Figure 2), all days falling into daytime and nighttime division are expected to have same outcome. Therefore, some days are selected as case study (see Figures 3 and 4).

The response is irrelevant. From the figure, I see that there is no correlation. You can calculate linear regression, but it is statistically insignificant. The regression reflects only your data set features instead of influence.  

3) I indicated: “Figures 3 and 4 also did not convince me. I am not sure that the effects are connected to lightning.”

The authors responded: “Figures 3 and 4 are specific examples showing the real impact of dominant day and nighttime lightning. In the bottom panels of each day, the ionosphere disturbances are revealed. ie Some hours after lightning in daytime and almost immediate in nighttime. The disturbances are linked to lightning as the selected days are free from geomagnetic storm and solar flare which are the main influences of the ionosphere.

Cross-correlation as suggested has been applied to the days in Figures 3 and 4. This has thrown more emphasis on  the time delay between lightning and ROTI. The coefficients are also positive indicating that the ROTI is affected by lightning.

Lightning interaction with ionosphere through gravitational waves, electrical discharges and now added Trimpi effect (see number 3 below) directly point to lightning as the source of disturbance.

These show that GNSS can provide more evidence to earlier studies which used instrumentation and data techniques like VLF -transmitter & receivers in describing lightning effects on ionosphere.

There are too many ways to increase ROTI, so find some increase (or decrease) is not a problem. Such different delays (up to 5 hours) show that the source are different. So, these figures still can not convince me.

4) I indicated: “I am not sure that lightning could produce such energy to influence significantly – I would wait for tinny effects in statistics or case studies. So, the authors should explain awaiting effects and mechanisms.”

The authors responded:

? “Trimpi effect, a mechanism which best describes effect of VLF from sources such as lightning on nighttime ionosphere has been incorporated. Trimpi effect gives a better explanation for the almost immediate impact of dominant nighttime lightning on the ionosphere.

But they do not say about energy. And how to distinguish the source from other those, what is the region, and so on.

5) I indicated: Referencing. Almost no pioneering papers were mentioned. My knowledge says that for cyclones pioneering papers are the papers by Polyakova, earthquakes – Calais, rocket launch – Afraimovich, detrending – Maletckii.

The authors responded nothing. They add some (!) references. I supposed that deep work on referencing should be done finding most relevant pioneering papers rather than just finding some papers of mentioned authors and adding them to the existing references.

6) I indicated: Finally, I feel that correlation is statistically insignificant. The paper needs more statistically grounding.

The authors responded nothing. I also do not see changes in the text with estimating statistical significance, a false positive/negative errors (lightning without ROTI and ROTI without lightning), distribution of delays.

Author Response

Response to Round 1 of Reviewer 2 comments to remotesensing-1757079

Dear Reviewer,

Once again thank you for your time, effort and knowledge sharing to improve the manuscript

The concerns and suggestions you raised are well received, addressed in the revised manuscript, and elaborated here.

  1. Clarity on “New ROTI”

The sampling rate of the GNSS receivers is 30s. ROTI according to its definition by Pi et al is computed at this rate over a running window of 5mins.

The ROTIavg (“New ROTI”) is just an average of the already calculated ROTI during every 30mins interval. That is, ROTI from time 00:00:00 to 00:30:00 are averaged, then 00:30:01 to 01:00:00 in that order. Therefore, the perturbations are still observed.

  1. “Non-convincing Figure 2”

As you clearly stated

“The regression reflects only your data set features instead of influence.”

We also share this idea by stating that “Correlation may not necessarily mean causation” (Line 213)

We then further look at the lightning interactions with the ionosphere in the discussion section. The physical mechanisms supported the observations we made in this current study and that of Amin et al.

We then used cross-correlation another statistical means which studies variables relative to each other based on time to derive new information. The results are shown in top right rows of Figures 3 and 4 (Line 260-270) and Figure 5 (Line 297-305).

The results from the use of cross-correlation best matches the physical mechanisms underlying lightning interactions with the ionosphere as compared to linear regression.

  1. There are too many ways to increase ROTI, so find some increase (or decrease) is not a problem. Such different delays (up to 5 hours) show that the source is different. So, these figures still can not convince me

From the selection criteria only days of lightning activity are chosen. The source of ROTI increase (or decrease) in this study would therefore be related to lightning as the days are free from geomagnetic storm and solar flare influences. The influence of gravity wave from lightning causing the delay as seen in top rows of Figures 3 and in the panels of Figure 5 are elaborated in Lines 208-216. The immediate impact of lightning influence for the evening time seen in top rows of Figure 4 are also elaborated in Lines 226-236.

  1. “I am not sure lightning could produce such energy”

From Discussion section of the manuscript EM and Trimpi effect arousing from lightning are elaborated to show their influence.

On source distinguishing, that would be a good area for further research with the use of GNSS data and techniques as to the best of our knowledge such distinction studies are yet to be undertaken.

  1. Referencing

Thank you for drawing our attention to this so that due credit is given to others for their good work done.

The relevant pioneering studies have been cited.

  1. “I feel that correlation is statistically insignificant.”

Our views on this are already stated in point 2 above.

What we derive is that in explaining activities of natural phenomena like lightning, different approaches in this case statistical means should be encouraged.

Reviewer 3 Report

I thank the authors for their answers and for taking into account many of my remarks.

Author Response

Response to Round 1 of Reviewer 3 comments to remotesensing-1757079

Dear Reviewer,

Thank you for your time, effort and knowledge sharing to improve the manuscript and further recommendation for it to be published

Your input was more than valuable, and it is highly appreciated.

Round 2

Reviewer 2 Report

Dear Editor and authors, 

While the main hypothesis is that lightnings influence the ionosphere (ROTI), it requires clear evidence. Correlation without statistical significance has no sense. Adding non-informative data caused by statistics could significantly change the results. See my figure in the att. Here the first panel shows a regression like those in Fig. 2 with negative correlation. However, it is just two different parts (see the second panel) caused by different mechanisms, and both are with positive regression. Excluding some data from "group 1" (see the third panel) make correlation coefficient almost zero. So, I still feel that figures 2 is not convincing to make the conclusions. Because the authors did not calculate statistical significance (and I think it is low) most probably the interpretation is not correct. About figs 3-4 I also have doubts that the correlations are statistically significant.

Author Response

Response to Round 2 of Reviewer 2 comments to remotesensing-1757079

Dear Reviewer,

Thank you for such elaborative response on the statistical correlation and its significance in your attachment

The concerns and suggestions you raised are well received, addressed in the revised manuscript, and elaborated here.

We have cross-checked our computations and provided the statistical significance values as suggested.

Such information is very vital, but it eluded us to add to the writeup as we were much fixated on the physical mechanisms

Please see lines 172-173 for that of linear correlation.

That of cross-correlation can be found in lines 347-367 And table 2.

Statistical significance value for cross-correlation is computed according to the information provided at Minitab

(https://support.minitab.com/en-us/minitab/18/help-and-how-to/modeling-statistics/time-series/how-to/cross-correlation/interpret-the-results/all-statistics-and-graphs/#:~:text=The%20interpretation%20for%20the%20cross,0.5547%20%3D%20the%20correlation%20is%20significant.)

This manuscript is a resubmission of an earlier submission. The following is a list of the peer review reports and author responses from that submission.

Round 1

Reviewer 1 Report

The paper presents really interesting study on the coupling of different geophysical phenomena – lightning activity and the ionospheric turbulence. The paper can be published in the present form. I just recommend authors to consider the following issue.

Typical intensity of electromagnetic wave generated by lightning (whistler) is too weak to produce any considerable modification of the ionosphere. The more efficient mechanism of the whistler impact on the ionosphere is Trimpi effect – the triggering of energetic electron precipitation by a whistler pulse. There is a lot of literature on this topic. This mechanism should be included into consideration.

Author Response

Dear Sir/Madam

Thanks for your time and efforts to improve this mansucript

Please find attached our responses and the revised mansucript

Reviewer 2 Report

The manuscript “Comparative study of Dominant Daytime and Nighttime Lightning occurrence and their impacts on Ionosphere Disturbances” is dedicated rather important studying of the ionosphere response to the tropospheric lightnings. The phenomena is valuable for modeling of upward impact to the ionosphere and can be taking in to account in the space weather forecasting. Usage the GNSS stations as a indicator of response is very interesting because the devices are widely distributed and in case of positive results may be valuable meaning for the forecasting. But some discrepancies prevent to publish the manuscript now.

First. The studied correlation of indexes “hourly lightning occurrence” and “hourly ROTI over threshold occurrence” seems invalid. Because measure of the scintillation is namely the value of variance (standard deviation, ROTI) rather than number of time intervals when the variance was over some threshold value. The value of ROTI itself is more exact index than "ROTI over threshold occurrence rate". Therefore one should use ROTI values for analysis without thresholding.

Second. The scatter plots presented on the left side of Figure 2 seems are not adopted for correlation analysis - the statistic is very poor and seems the relation is not linear. This arise when the compared correlating raws are shifted in time (see Figures 3 and 4). So one need to analyze the cross-correlation function here rather than simple correlation coefficient.

The work should be remade dropping incorrect indexes for comparative analysis and removing the wrong claims based on them (e.g. line 56 :"The reason could be attributed to lightning interactions with the ionosphere in the different geographical regions."). Or supply presented results by corresponding reliable explanation.

Some extra irregularities found:

Figure 1 Better will be put the same color scale for all panels.

Line 124 Change «Moreso» to «More so»\

lines 127-129 «This difference in threshold brings to bear an important factor to be considered when regional and global models are being developed.» What namely the factor should be taken in to account?

Based on this I ask the major revision of manuscript.

Author Response

Dear Sir/Madam

Thanks for your efforts and suggestions to improve this manuscript.

Please find attached the revised manuscript

Reviewer 3 Report

I would say that the paper tries to study the fine effects of how lightning could influence the Earth's ionosphere. However, the results are not convincing. I am not sure that the authors obtain a real effect rather than an experiment design artefact (intention-to-treat error). For such a research accurate statistical test is required as well as a deep analyzing the results.

1) The authors introduced “new ROTI” that differs significantly from those suggested by Pi et al. Pi et al used 30-sec slant TEC data, while here 5-min vertical TEC data is used. I do not think that at such a low resolution this “new ROTI” would correlate with ionospheric scintillation (as in [21,22]) due to different time scales. According to Jacobsen, even different sampling rates could result in different effects (DOI: 10.1051/swsc/2014031).

2) Figure 2 is so unconvincing. I would say about no correlation. There are different ROTI occurrences at low lightning occurrence (which occur more often) and constant ROTI at the different (higher) lightning occurrences. From this figure, I would conclude that lightning does not influence ROTI.

3) Figures 3 and 4 also did not convince me. I am not sure that the effects are connected to lightning.

4) I am not sure that lightning could produce such energy to influence significantly – I would wait for tinny effects in statistics or case studies. So, the authors should explain awaiting effects and mechanisms.

5) Referencing. Almost no pioneering papers were mentioned. My knowledge says that for cyclones pioneering papers are the papers by Polyakova, earthquakes – Calais, rocket launch – Afraimovich, detrending – Maletckii.

6) Finally, I feel that correlation is statistically insignificant. The paper needs more statistically grounding.

Minor remarks:

1) ROTI is a root-mean-square of the TEC rate.

2) ROT1 -> ROTI.

3) No of -> Number of

4) Explain sfu (solar flux units + 1 sfu value)

Author Response

Dear Reviewer,

Thank you for your efforts and comments to improve the manuscript.

Please find attached the revised version

Reviewer 4 Report

This is the review of the paper by Osei-Poku et al. entitled "Comparative study of Dominant Daytime and Nighttime Lightning occurrence and their impacts on Ionosphere Disturbances" submitted for publication in the journal Remote Sensing.

This paper shows a statistical analysis on a series of TEC variation measurements in relation to thunderstorm activity over Hong Kong during four years between 2014 and 2017. The authors calculate a parameter accounting for the 30-minute variability of ionospheric disturbances, the ROTI. The hourly average distribution of ROTI and lightning occurrence are calculated per year for the four years of measurements. The authors find a negative correlation between the two distributions, in contrast to a previous study in South Africa on case studies. The authors suggest two mechanisms that could lead to a positive correlation between the two distributions: gravity wave emission and ionization by the interaction of electromagnetic waves from lightning and the lower ionosphere. Examples of individual cases, during the day and at night, are proposed to support their assumptions. The authors conclude, without further calculations, that taking into account these two assumptions the two distributions are finally positively correlated.

The discussion of the two hypotheses is strongly lacking in numerical values (e.g. the average vertical velocity of a gravity wave, order of magnitude of the electron density perturbations relative to the measured TEC changes...), includes inaccuracies in the terms used, and above all the final conclusion that the two distributions are finally positively correlated is purely unsupported by calculations (correlation coefficients are given with no explanation of the calculation of the change in the hourly distributions). More details are given in the specific comments below.

For these reasons but also because the data are of interest, I recommend a major revision.

Specific comments

Lines 25-26: it is not the lightning that generates the gravity waves but the convection in the cumulonimbus, in other words by the thunderstorms themselves.

I will come back to this later.

Line 36: Give a reference for rocket launch measurements

Line 38: Remove "Thunderstorm/lightning," it is a repetition with the end of the sentence.

Line 44: Blanc et al. (2018) also showed gravity wave measurements in the ionosphere following thunderstorms for the European area. Blanc, E., Ceranna, L., Hauchecorne, A., et al. Toward an Improved Representation of Middle Atmospheric Dynamics Thanks to the ARISE Project. Surv Geophys 39, 171-225 (2018). https://doi.org/10.1007/s10712-017-9444-0

Lines 58-59: The sentence "The results are compared to Amin [15] and those in other geographical regions and discussed." should be moved line 54 after "effective comparison."

Lines 59-61: This sentence concluding the introduction would be better placed in the conclusion of the article.

Line 66: What do you mean explicitly by "Total current"? The sum of the current of each flash? If so, how do you measure or calculate it?

Line 71: There seem to be 19 stations in Hong Kong and 3 in Macau. Did you use the data from only one of these receivers, from several, from all the ones available? Please clarify this point. 

Line 74: "(accessed on 14 June 2019)", it is surprising that the authors did not check more recently that the data is still accessible. It appears that the website is still working.

Line 79: You can cut the sentence after "TEC variance".

Line 81: The sentence starting with "At 15° elevation ..." is not clear. I suggest: "A 15º elevation cut off angle is used to eliminate multipath effect [6].

Line 91: Do you mean ROTI (instead of ROT1)?

Line 92: remove the "s" from Equation.

Lines 97-98: I suggest cutting the sentence in two parts. "ROTI as shown in Equation 2 is usually computed for a single satellite-receiver pair; a threshold set at 0.2TECu [24] indicates ionosphere scintillation occurrence."

Lines 98-99: I also propose to reword the sentence as follows: "However, ROTI average (ROTIavg) is the ROTI average value over 30 minutes for all satellites received by a single station; after Oladipo, et al. [25], a scintillation occurrence is detected for a threshold of 0.8TECu." Can you confirm that the calculation is for a single station only?

Line 106: replace "found" with "looked for"

Line 112: I think table 1 represents the number of days with lightning and when geomagnetic activity is low, right? If so, specify in the text and in the legend of Table 1.

Line 117: from the point of view of paper organization, I will separate the results from the discussion.

Line 118-129: you need to reorganize the ideas in this paragraph. First the findings and then the treatment hypotheses. Here is a suggestion:

"Figure 1 shows ROTIavg for 2014 to 2017. From Figure 1, the highest values are mostly in the nighttime and between 0.07 - 0.2TECu, this indicates that scintillation is often a nighttime event. This is coherent with Tang, et al [23] and Ji, et al [18] which have shown that greater percentage of ROTI lies between 0.02 - 0.05TECu in the Hong Kong region. In this present study, ROTIavg threshold is set at 0.075TECu instead of 0.8TECu, differently to Oladipo, et al. [25], but as Nishioka, et al. [28] who suggested scintillation threshold should be at this value in the Asian region. Moreso, the years 2014-2017 are at the declining phase of the 24th solar cycle [29] where scintillation values are low compared to the solar active years of 2002 and 2012 in Oladipo, et al. [25] and Amin [15] respectively. This difference in threshold brings to bear an important factor to be considered when regional and global models are being developed."

Furthermore, you could better specify the variability of ROTIavg values with the following statistical parameters: mean, median, 25th and 75th percentile, ...

Figure 1: you can't see much on the figure. The colorscale should be better adapted, a low threshold should be set equal to the detection threshold, the maximum values should be saturated at 0.8 TECu/minute (no red or orange dots are seen).

Lines 131-133: add "axis" after "abscissa" and "ordinate" or replace by "x-axis" and "y-axis" respectively. The unit of the color bar is TECu/minute.

Line 137: specify for each year the peak time of each distribution.

Lines 143-145: remove the capital letters from Lightning. Clarify that these are average values. The last sentence does not have a verb.

Line 146: add the beginning of the paragraph "4. Discussion" here.

Line 147: replace "onus" with "objective" or "aim".

Lines 157-170: the diurnal variation of thunderstorm activity whatever the latitude (at least between +/- 60°) and longitude is now well established thanks to the global networks of localization of lightning on the ground (such as WWLLN or GLD360) or optical space measurements by the OTD, LIS or ISS-LIS experiments for more than twenty years. There is a maximum around 14 or 15 hours TL and a minimum around 10 hours TL. Such a long paragraph is therefore not necessarily needed. Especially the sentence "A study on a global scale would be needed to provide more evidence to this observation." is inappropriate.

Lines 170-172: The sentence "Statistically, lightning and ROTIavg peaks coinciding at the same time stand a chance of giving a positive correlation value while peaks which occur at different times would give a negative correlation value." is not appropriate.

Lines182 179: reword these sentences: "Lightning may affect the electron density in lower ionosphere (particularly the D-layer) through two basic physical processes: generating gravity wave (GW) [37] and electromagnetic wave (EM) [16]. Although GW is a transversal wave, any changes of its amplitude generated from a lightning discharge will perform as a longitudinal wave that propagates upwards at the speed of order of sound in air." It is inaccurate. It is not lightning that causes gravity waves but the convection present in thunderstorm clouds. The gravity wave will generate a disturbance of the mesosphere (60-90 km) and the lower thermosphere (above 90 km), thus of the neutral atmosphere, which drags electrons from the ionosphere.

Line 182-183: What do you mean by "On the other side, the ionospheric D-layer is lower and thicker during daytime than nighttime"? From the point of view of electron concentration, the D-layer is denser during daytime than at night, since the latter disappears in the absence of solar radiation. Moreover, in relation to the VTEC calculation, one must remember the orders of magnitude of electron density between the D layer and the F layer, even at night. The D-layer accounts for very little in the value of the VTEC.

Line 183: "the lightning-generated GW" incorrect as described above.

Line 184: specifying the propagation time a priori "some hours" is elusive. Also, specify the vertical speed range of the gravity waves ("its very low propagation speed at a high altitude"). Without this, it is not possible to determine even if these wave behaviours are compatible with a simple calculation of the speed deduiced from time lag between the distyribution; if one assumes a 7 to 9 hours gap between the peak of the distributions and a propagation over 350 km height (from the ground to the reference altitude in the ionosphere) one finds a vertical speed between 10 and 15 m/s.

Lines 197-200: can you specify the range of altitude in which increases and decreases of electron density are calculated or measured after the emission of the EMP of a flash? Moreover, and especially, it is known that these disturbances last only a few seconds to tens of seconds. What impact do you think possible on the ROTIavg knowing that we have here an average over 30 minutes? Moreover, what is the order of magnitude of these  electron denity changes in absolute value and especially in relative value with respect to the electron density of the F layer? It seems to me that all this is very small and difficult to measure in the TEC. In any case you have to do this calculation to show the contrary.

Line 203-204: "Thus, lightning which occur dominantly in the evening sees the ionosphere disturbance almost instantly leading to a positive correlation." This statement needs to be verified! The realization of distribution for only the night measurements (18-6LT) or day measurements (6-18 LT) on the 4 years like what is done on Figure 2 should make it possible to show it if that indeed exists.

Lines 205-207: why go directly to the case study before having a statistical study isolating two sectors: day and night. Also, I don't see why you divide the night sector (00-07 and 18-23) into two. Moreover, calling the sector (00-07) dawn is very curious.

Line 213: why choose such a narrow and low frequency filter for acoustic signals? We are at the lower limit of the acoustic spectral range. If the waves come from thunder, the latter preferentially emit waves above 1Hz (you can refer for example to the articles cited on this subject by Blanc et al. (2018) whose reference was given above).

Line 216: what do you mean by "Figures 3 and 4 show the days, the available PRN ...".

Line 239-242: how do you explain that the ROTI peak precedes the lightning peak in Figure 4g? Would the propagation be non-linear?

Line 245: "would yield positive correlation values of 0.74, 0.85, 0.82 and 0.8 for 2014 to 2017" How do you get these correlation results without doing a gravity wave propagation calculation or anything else? These values without further explanation are unacceptable as such. Moreover, they are used for the end of the paper as proof that the data are finally correlated.

Line 247: "This now agrees that the diurnal hourly lightning and associated ROTI occurrence have a positive correlation." Delete this sentence (this is gratuitous statement) or justify it with a minimum of calculations.

Lines 252-264: I do not agree with your conclusion as it is not demonstrated. No calculations, even with orders of magnitude, support your assumptions.

Line 394: add spaces in "sporadicEby

Author Response

Dear Reviewer,

Thank you for your wonderful constructive criticisms to improve the manuscript.

Please find attached the revised version

Round 2

Reviewer 2 Report

In spite of multiple mention of the "occurence rate" the method should be very sensitive to the temporal resolution of and to the threshold adjustement. For the same seqiuence minute temporal resolution and seconds temporal resolution yelds occurence rate 60 per hour and 3600 per hour  correspondingly if the threshould will be low. 

Reviewer 4 Report

I thank the authors for their answer and for taking into account many of my remarks. Nevertheless, there are still some questions that I had asked concerning the discussion and which remained without answer or modification in the text and whose answer is essential for the validation of the hypotheses suggested by the authors. I therefore take the liberty of asking them a second time.

I will not make line by line corrections here because the pdf version of the modified text that I received mixes the deleted and added parts (text or figure) without differentiation, making it difficult to read. I will note however that "gravitational wave" should be replaced by "gravity wave".

First of all, I insist that additional studies (line 204-205) on the Local Time distribution of lightning occurrence do not need to be done since this already exists, e.g. Chronis and Koshack, BAMS, 2017.

It is essential that you provide orders of magnitude of the phenomena that you believe would correct for the temporal discrepancy in maximum lightning occurrence and ionospheric disturbances.

  1. Gravity wave process. From the cross-correlation calculations that you present (Figure 5), you can specify (line 231) the temporal gap between the two distributions. This 7 to 9 hours gap implies an average vertical velocity of gravity waves between the ground and the ionosphere (350 km) of 10 to 15 m/s. Can you give typical values of this vertical velocity associated with at least one reference? This will strengthen the likelihood that your propagation delay hypothesis is reasonable.
  2. Concerning the ionization induced in the D region by the lightning EMP or the Trimpi effect, it is important that you quantify this perturbation in an absolute and relative way with respect to the VTEC. This will verify that the perturbations you observe are consistent with the phenomenon you believe is the source of these perturbations. Finally, since this induced ionization lasts only a few seconds for each flash, how is it possible that it can be measured in GNSS signals averaged over 30 minutes (ROTIavg)? You must also provide an answer to this question.

Finally, in your conclusions, you should remain more moderate by indicating that this temporal discrepancy could be explained by one or the other (or both) of these two processes and that it would deserve simulations to confirm their likelihood.